# Compositional Visual Generation with Energy Based Models

**Yilun Du**
MIT CSAIL
yilundu@mit.edu

**Shuang Li**
MIT CSAIL
lishuang@mit.edu

**Igor Mordatch**
Google Brain
imordatch@google.com

## Abstract

A vital aspect of human intelligence is the ability to compose increasingly complex concepts out of simpler ideas, enabling both rapid learning and adaptation of knowledge. In this paper we show that energy-based models can exhibit this ability by directly combining probability distributions. Samples from the combined distribution correspond to compositions of concepts. For example, given one distribution for smiling face images, and another for male faces, we can combine them to generate smiling male faces. This allows us to generate natural images that simultaneously satisfy conjunctions, disjunctions, and negations of concepts. We evaluate compositional generation abilities of our model on the CelebA dataset of natural faces and synthetic 3D scene images. We showcase the breadth of unique capabilities of our model, such as the ability to continually learn and incorporate new concepts, or infer compositions of concept properties underlying an image.

## 1 Introduction

Humans are able to rapidly learn new concepts and continuously integrate them among prior knowledge. The core component in enabling this is the ability to compose increasingly complex concepts out of simpler ones as well as recombining and reusing concepts in novel ways [5]. By combining a finite number of primitive components, humans can create an exponential number of new concepts, and use them to rapidly explain current and past experiences [16]. We are interested in enabling such capabilities in machine learning systems, particularly in the context of generative modeling.

Past efforts have attempted to enable compositionality in several ways. One approach decomposes data into disentangled factors of variation and situate each datapoint in the resulting - typically continuous - factor vector space [29, 9]. The factors can either be explicitly provided or learned in an unsupervised manner. In both cases, however, the dimensionality of the factor vector space is fixed and defined prior to training. This makes it difficult to introduce new factors of variation, which may be necessary to explain new data, or to taxonomize past data in new ways. Another approach to incorporate the compositionality is to spatially decompose an image into a collection of objects, each object slot occupying some pixels of the image defined by a segmentation mask [28, 6]. Such approaches can generate visual scenes with multiple objects, but may have difficulty in generating interactions between objects. These two incorporations of compositionality are considered distinct, with very different underlying implementations.

In this work[*], we propose to implement the compositionality via energy based models (EBMs). Instead of an explicit vector of factors that is input to a generator function, or object slots that are blended to form an image, our unified treatment defines factors of variation and object slots via energy functions. Each factor is represented by an individual scalar energy function that takes as input an image and outputs a low energy value if the factor is exhibited in the image. Images that exhibit the

---

[*]Code and data available at `https://energy-based-model.github.io/compositional-generation-inference/`

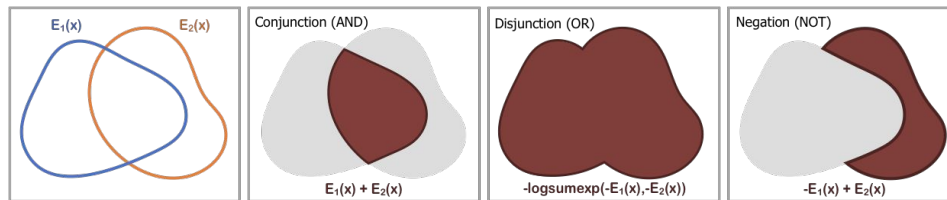

Figure 1: Illustration of logical composition operators over energy functions $E_1$ and $E_2$ (drawn as level sets where red = valid areas of samples, grey = invalid areas of samples).

factor can then be generated implicitly through an Markov Chain Monte Carlo (MCMC) sampling process that minimizes the energy. Importantly, it is also possible to run MCMC process on some *combination* of energy functions to generate images that exhibit multiple factors or multiple objects, in a globally coherent manner.

There are several ways to combine energy functions. One can add or multiply distributions as in mixtures [25, 6] or products [11] of experts. We view these as probabilistic instances of logical operators over concepts. Instead of using only one, we consider three operators: logical conjunction, disjunction, and negation (illustrated in Figure 1). We can then flexibly and recursively combine multiple energy functions via these operators. More complex operators (such as implication) can be formed out of our base operators.

EBMs with such composition operations enable a breadth of new capabilities - among them is a unique approach to continual learning. Our formulation defines concepts or factors implicitly via examples, rather than pre-declaring an explicit latent space ahead of time. For example, we can create an EBM for concept "black hair" from a dataset of face images that share this concept. New concepts (or factors), such as hair color can be learned by simply adding a new energy function and can then be combined with energies for previously trained concepts. This process can repeat continually. This view of few-shot concept learning and generation is similar to work of [23], with the distinction that instead of learning to generate holistic images from few examples, we learn *factors* from examples, which can be composed with other factors. A related advantage is that finely controllable image generation can be achieved by specifying the desired image via a collection of logical clauses, with applications to neural scene rendering [4].

Our contributions are as follows: first, while composition of energy-based models has been proposed in abstract settings before [11], we show that it can be used to generate plausible natural images. Second, we propose a principled approach to combine independent trained energy models based on logical operators which can be chained recursively, allowing controllable generation based on a collection of logical clauses at test time. Third, by being able to recursively combine independent models, we show our approach allows us to extrapolate to new concept combinations, continually incorporate new visual concepts for generation, and infer concept properties compositionally.

## 2 Related Work

Our work draws on results in energy based models - see [17] for a comprehensive review. A number of methods have been used for inference and sampling in EBMs, from Gibbs Sampling [12], Langevin Dynamics [31, 3], Path Integral methods [2] and learned samplers [13, 26]. In this work, we apply EBMs to the task of compositional generation.

Compositionality has been incorporated in representation learning (see [1] for a summary) and generative modeling. One approach to compositionality has focused on learning disentangled factors of variation [8, 15, 29]. Such an approach allows for the combination of existing factors, but does not allow the addition of new factors. A different approach to compositionality includes learning various different pixel/segmentation masks for each concept [6, 7]. However such a factorization may have difficulty capturing the global structure of an image, and in many cases different concepts cannot be explicitly factored using attention masks.

In contrast, our approach towards compositionality focuses on composing separate learned probability distribution of concepts. Such an approach allows viewing factors of variation as constraints [19]. In prior work, [10] show that products of EBMs can be used to decompose complex generative modeling problems to simpler ones. [29] further apply products of distributions over the latent space of VAE to define compositions. [9] show that additional compositions in VAE latent space. Both of them rely on joint training to learn compositions of a fixed number of concepts. In contrast,

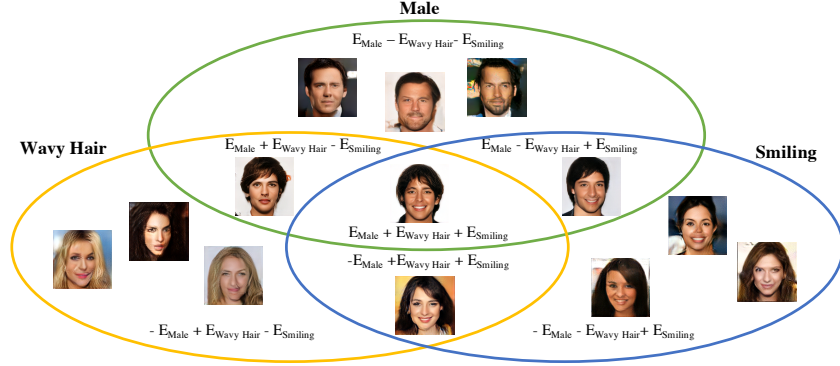

Figure 2: Concept conjunction and negation. All the images are generated through the conjunction and negation of energy functions. For example, the image in the central part is the conjunction of male, black hair, and smiling energy functions. Equations for composition explained in page 4.

in this work, we show how we can realize concept compositions using completely **independently** trained probability distributions. Furthermore, we introduce three compositional logical operators of conjunction, disjunction and negation can be realized and nested together through manipulation of independent probability distributions of each concept.

Our compositional approach is inspired by the goal of continual lifelong learning - see [20] for a thorough review. New concepts can be composed with past concepts by combining new independent probability distributions. Many methods in continual learning are focused on how to overcome catashtophic forgetting [14, 18], but do not support dynamically growing capacity. Progressive growing of the models [24] has been considered, but is implemented at the level of the model architecture, whereas our method composes independent models together.

## 3    Method

In this section, we first give an overview of the Energy-Based Model formulation we use and introduce three logical operators over these models. We then discuss the unique properties such a form of compositionality enables.

### 3.1    Energy Based Models

EBMs represent data by learning an unnormalized probability distribution across the data. For each data point $\mathbf{x}$, an energy function $E_\theta(\mathbf{x})$, parameterized by a neural network, outputs a scalar real energy such that the model distribution

$$p_\theta(x) \propto e^{-E_\theta(x)}. \tag{1}$$

To train an EBM on a data distribution $p_D$, we use contrastive divergence [10]. In particular we use the methodology defined in [3], where a Monte Carlo estimate (Equation 2) of maximum likelihood $\mathcal{L}$ is minimized with the following gradient

$$\nabla_\theta \mathcal{L} = \mathbb{E}_{x^+ \sim p_D} \nabla_\theta E_\theta(x^+) - \mathbb{E}_{x^- \sim p_\theta} \nabla_\theta E_\theta(x^-). \tag{2}$$

To sample $x^-$ from $p_\theta$ for both training and generation, we use MCMC based off Langevin dynamics [30]. Samples are initialized from uniform random noise and are iteratively refined using

$$\tilde{\mathbf{x}}^k = \tilde{\mathbf{x}}^{k-1} - \frac{\lambda}{2} \nabla_\mathbf{x} E_\theta(\tilde{\mathbf{x}}^{k-1}) + \omega^k, \ \omega^k \sim \mathcal{N}(0, \lambda), \tag{3}$$

where $k$ is the $k^{th}$ iteration step and $\lambda$ is the step size. We refer to each iteration of Langevin dynamics as a negative sampling step. We note that this form of sampling allows us to use the gradient of the combined distribution to generate samples from distributions composed of $p_\theta$ and the other distributions. We use this ability to generate from multiple different compositions of distributions.

### 3.2    Composition of Energy-Based Models

We next present different ways that EBMs can compose. We consider a set of independently trained EBMs, $E(\mathbf{x}|c_1), E(\mathbf{x}|c_2), \ldots, E(\mathbf{x}|c_n)$, which are learned conditional distributions on underlying concept codes $c_i$. Latent codes we consider include position, size, color, gender, hair style, and age, which we also refer to as concepts. Figure 2 shows three concepts and their combinations on the CelebA face dataset and attributes.

**Concept Conjunction** In concept conjunction, given separate independent concepts (such as a particular gender, hair style, or facial expression), we wish to construct an output with the specified gender, hair style, and facial expression – the combination of each concept. Since the likelihood of an output given a set of specific concepts is equal to the product of the likelihood of each individual concept, we have Equation 4, which is also known as the product of experts [11]:

$$p(x|c_1 \text{ and } c_2, \ldots, \text{ and } c_i) = \prod_i p(x|c_i) \propto e^{-\sum_i E(x|c_i)}. \tag{4}$$

We can thus apply Equation 3 to the distribution that is the sum of the energies of each concept. We sample from this distribution using Equation 5 to sample from the joint concept space with $\omega^k \sim \mathcal{N}(0, \lambda)$.

$$\tilde{\mathbf{x}}^k = \tilde{\mathbf{x}}^{k-1} - \frac{\lambda}{2} \nabla_{\mathbf{x}} \sum_i E_\theta(\tilde{\mathbf{x}}^{k-1}|c_i) + \omega^k. \tag{5}$$

**Concept Disjunction** In concept disjunction, given separate concepts such as the colors red and blue, we wish to construct an output that is either red or blue. This requires a distribution that has probability mass when any chosen concept is true. A natural choice of such a distribution is the sum of the likelihood of each concept:

$$p(x|c_1 \text{ or } c_2, \ldots \text{ or } c_i) \propto \sum_i p(x|c_i)/Z(c_i). \tag{6}$$

where $Z(c_i)$ denotes the partition function for each concept. A tractable simplification becomes available if we assume all partition functions $Z(c_i)$ to be equal

$$\sum_i p(x|c_i) \propto \sum_i e^{-E(x|c_i)} = e^{\text{logsumexp}(-E(x|c_1), -E(x|c_2), \ldots, -E(x|c_i))}, \tag{7}$$

where $\text{logsumexp}(f_1, \ldots, f_N) = \log \sum_i \exp(f_i)$. We can thus apply Equation 3 to the distribution that is a negative smooth minimum of the energies of each concept to obtain Equation 8 to sample from the disjunction concept space:

$$\tilde{\mathbf{x}}^k = \tilde{\mathbf{x}}^{k-1} - \frac{\lambda}{2} \nabla_{\mathbf{x}} \text{logsumexp}(-E(x|c_1), -E(x|c_2), \ldots, -E(x|c_i)) + \omega^k, \tag{8}$$

where $\omega^k \sim \mathcal{N}(0, \lambda)$. While the assumption that leads to Equation 7 is not guaranteed to hold in general, in our experiments we empirically found the partition function $Z(c_i)$ estimates to be similar across partition functions (see Appendix) and also analyze cases in which partitions functions are different in the Appendix. Furthermore, the resulting generation results do exhibit equal distribution across disjunction constituents in practice as seen in Table 1.

**Concept Negation** In concept negation, we wish to generate an output that does not contain the concept. Given a color red, we want an output that is of a different color, such as blue. Thus, we want to construct a distribution that places high likelihood to data that is outside a given concept. One choice is a distribution inversely proportional to the concept. Importantly, negation must be defined with respect to another concept to be useful. The opposite of alive may be dead, but not inanimate. Negation without a data distribution is not integrable and leads to a generation of chaotic textures which, while satisfying absence of a concept, is not desirable. Thus in our experiments with negation we combine it with another concept to ground the negation and obtain an integrable distribution:

$$p(x|\text{not}(c_1), c_2) \propto \frac{p(x|c_2)}{p(x|c_1)^\alpha} \propto e^{\alpha E(x|c_1) - E(x|c_2)}. \tag{9}$$

We found the smoothing parameter $\alpha$ to be a useful regularizer (when $\alpha = 0$ we arrive at uniform distribution) and we use $\alpha = 0.01$ in our experiments. The above equation allows us to apply Langevin dynamics to obtain Equation 10 to sample concept negations.

$$\tilde{\mathbf{x}}^k = \tilde{\mathbf{x}}^{k-1} - \frac{\lambda}{2} \nabla_{\mathbf{x}} (\alpha E(x|c_1) - E(x|c_2)) + \omega^k, \tag{10}$$

where $\omega^k \sim \mathcal{N}(0, \lambda)$.

**Recursive Concept Combinations** We have defined the three classical symbolic operators for concept combinations. These symbolic operators can further be recursively chained on top of each to specify more complex logical operators at test time. To our knowledge, our approach is the only approach enabling such compositionality across independently trained models.

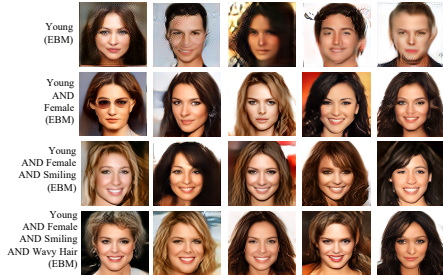

Figure 3: Combinations of different attributes on CelebA via concept conjunction. Each row adds an additional energy function. Images on the first row are conditioned on young, while images on the last row are conditioned on young, female, smiling, and wavy hair.

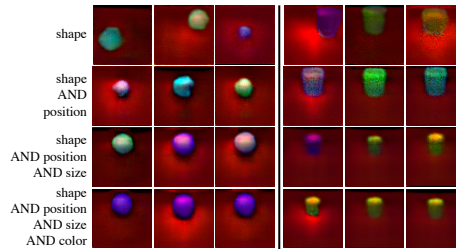

Figure 4: Combinations of different attributes on MuJoCo via concept conjunction. Each row adds an additional energy function. Images on the first row are only conditioned on shape, while images on the last row are conditioned on shape, position, size, and color. The left part is the generation of a sphere shape and the right is a cylinder.

# 4 Experiments

We perform empirical studies to answer the following questions: (1) Can EBMs exhibit concept compositionality (such as concept negation, conjunction, and disjunction) in generating images? (2) Can we take advantage of concept combinations to learn new concepts in a continual manner? (3) Does explicit factor decomposition enable generalization to novel combinations of factors? (4) Can we perform concept inference across multiple inputs?

In the appendix, we further show that approach enables better generalization to novel combinations of factors by learning explicit factor decompositions.

## 4.1 Setup

We perform experiments on 64x64 object scenes rendered in MuJoCo [27] (MuJoCo Scenes) and the 128x128 CelebA dataset. For MuJoCo Scene images, we generate a central object of shape either sphere, cylinder, or box of varying size and color at different positions, with some number of (specified) additional background objects. Images are generated with varying lighting and objects.

We use the ImageNet32x32 architecture and ImageNet128x128 architecture from [3] with the Swish activation [22] on MuJoCo and CelebA datasets. Models are trained on MuJoCo datasets for up to 1 day on 1 GPU and for 1 day on 8 GPUs for CelebA. More training details and model architecture can be found in the appendix.

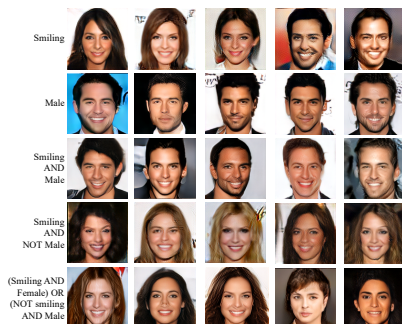

Figure 5: Examples of recursive compositions of disjunction, conjunction, and negation on the CelebA dataset.

| Model | Pos Acc | Color Acc |
|---|---|---|
| Color | 0.128 | 0.997 |
| Pos | 0.984 | 0.201 |
| Pos & Color | 0.801 | 0.8125 |
| Pos & ($\neg$ Color) | 0.872 | 0.096 |
| ($\neg$ Pos) & Color | 0.033 | 0.971 |
| Color [29] | 0.132 | 0.333 |
| Pos [29] | 0.146 | 0.202 |
| Pos & Color [29] | 0.151 | 0.342 |

| Model | Pos 1 Acc | Position 2 Acc |
|---|---|---|
| Pos 1 | 0.875 | 0.0 |
| Pos 2 | 0.0 | 0.817 |
| Pos 1 \| Pos 2 | 0.432 | 0.413 |

| Model | Pos/Color 1 Acc | Pos 2/Color 2 Acc |
|---|---|---|
| Pos 1 & Color 1 | 0.460 | 0.0 |
| Pos 2 & Color 2 | 0.0 | 0.577 |
| (Pos 1 & Color 1) \| (Pos 2 & Color 2) | 0.210 | 0.217 |

Table 1: Quantitative evaluation of conjunction (&), disjunction (|) and negation ($\neg$) generations on the Mujoco Scenes dataset using an EBM or the approach in [29]. Position = Pos. Each individual attribute (Color or Position ) generation is a individual EBM. (Acc: accuracy) Standard error is close to 0.01 for all models.

## 4.2 Compositional Generation

**Quantitative evaluation.** We first evaluate compositionality operations of EBMs in Section 3.2. To quantitatively evaluate generation, we use the MuJoCo Scenes dataset. We train a supervised classifier to predict the object position and color on the MuJoCo Scenes dataset. Our classifier obtains 99.3% accuracy for position and 99.9% for color on the test set. We also train seperate conditional EBMs on the concepts of position and color. For a given positional generation then, if the predicted position (obtained from a supervised classifier on generated images) and original conditioned generation position is smaller than 0.4, then a generation is consider correct. A color generation is correct if the predicted color is the same as the conditioned generation color.

In Table 1, we quantitatively evaluate the quality of generated images given combinations of conjunction, disjunction, and negation on the color and position concepts. When using either Color or Position EBMs, the respective accuracy is high. Conjunction(Position, Color) has high position and color accuracies which demonstrates that an EBM can combine different concepts. Under Conjunction(Position, Negation(Color)), the color accuracy drops to below that of Color EBM. This means negating a concept reduces the likelihood of the concept. The same conclusion for Conjunction(Negation(Position), Color). We compare with the approach in [29], using the author's online github repo, and find it produces blurrier and worse results.

To evaluate disjunction, we set Position 1 to be a random point in the bottom left corner of a grid and Position 2 to be a random point in the top right corner of a grid. The average results over 1000 generated images are reported in Table 1. Position 1 EBM or Position 2 EBM can obtain high accuracy in predicting their own positions. Disjunction(Position 1, Position 2) EBM generate images that are roughly evenly distributed between Position 1 and Position 2, indicating the disjunction can combine concepts additively. This trend further holds with conjunction, with Disjunction(Conjunction(Position 1, Color 1),Conjunction(Position 2, Color 2)) also being evenly distributed.

We further investigate implication using a composition of conjunctions and negations in EBMs. We consider the term (Position 1 AND (NOT Color 1)) AND ... AND (Position 1 AND (NOT Color 4)), which implicates Color 5. We find that are generations obtain 0.982 accuracy for Color 5.

**Qualitative evaluation.** We further provide qualitative visualizations of conjunction, disjunction, and negation operations on both MuJoCo Scenes and CelebA datasets.

*Concept Conjunction:* In Figure 3, we show the conjunction of EBMs is able to combine multiple independent concepts, such as age, gender, smile, and wavy hair, and get more precise generations with each energy models. Our composed generations obtain a FID of 45.3, compared to an FID of 64.5 of an SNGAN model trained on data conditioned on all four attributes. Our generations are also significantly more diverse than that of GAN model (average pixel MSE of 64.5 compared to 55.4 of the GAN model). Similarly, EBMs can combine independent concepts of shape, position, size, and color to get more precise generations in Figure 4. We also show results of conjunction with other logical operators in Figure 5.

*Concept Negation:* In Figure 5, row 4 shows images that are opposite to the trained concept using negation operation. Since concept negation operation should accompany with another concept as described in Section 3.2, we use "smiling" as the second concept. The images in row 4 shows the negation of male AND smiling is smiling female. This can further be combined with disjunction in the row 5 to make either "non-smiling male" or "smiling female".

*Concept Disjunction:* The last row of Figure 5 shows EBMs can combine concepts additively (generate images that are concept A or concept B). By constructing sampling using logsumexp, EBMs can sample an image that is "not smiling male" or "smiling female", where both "not smiling male" and "smiling female" are specified through the conjunction of energy models of the two concepts.

*Multiple object combination:* We show that our composition operations not only combine object concepts or attributes, but also on the object level. To verify this, we constructed a dataset with one green cube and a large amount background clutter objects (which are not green) in the scene. We train a conditional EBM (conditioned on position) on the dataset. Figure 7 "cube 1" and "cube 2" are the generated images conditioned on different positions. We perform the conjunction operation on the EBMs of "cube 1" and "cube 2" and use the combined energy model to generate images (row 3). We find that adding two conditional EBMs allows us to selectively generate two different cubes. Furthermore, such generation satisfies the constraints of the dataset. For example, when two

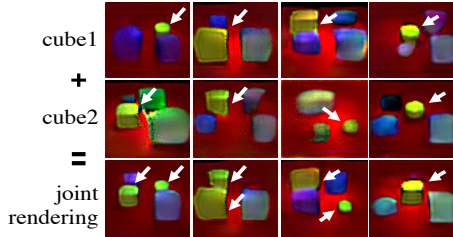
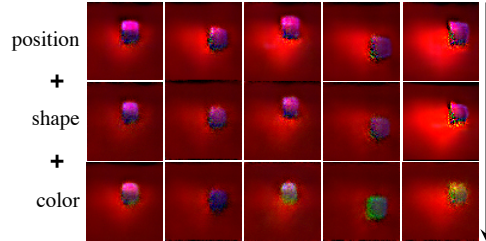

cube1

+

cube2

=

joint rendering

position

+

shape

+

color

Figure 7: Multi-object compositionality with EBMs. An EBM is trained to generate a green cube at location in a scene alongside other objects. At test time, we sample from the conjunction of two EBMs conditioned on different positions and sizes (cube 1 and 2) and generates cubes at both locations. Two cubes are merged into one if they are too close (last column).

Figure 8: Continual learning of concepts. A position EBM is first trained on one shape (cube) of one color (purple) at different positions (first row). A shape EBM is then trained on different shapes of one fixed color (purple) (second row). Finally, a color EBM is trained on shapes of many colors (third row). EBMs learn to combine concepts to many shapes (cube, sphere), colors and positions.

conditional cubes are too close, the conditionals EBMs are able to default and just generate one cube like the last image in row 3.

## 4.3 Continual Learning

We evaluate to what extent compositionality in EBMs enables continual learning of new concepts and their combination with previously learned concepts. If we create an EBM for a novel concept, can it be combined with previous EBMs that have never observed this concept in their training data? And can we continually repeat this process?

To evaluate this, we use the following methodology on MuJoCo dataset: 1) We first train a position EBM on a dataset of varying positions, but a fixed color and a fixed shape. In experiment, we use shape "cube" and color "purple". The position EBM allows us generate a purple cube at various positions. (Figure 8 row 1). 2) Next we train a shape EBM by training the model in combination with the position EBM to generate images of different shapes at different positions, but without training position EBM. As shown in Figure 8 row 2, after combining the position and shape EBMs, the "sphere" is placed in the same position as "cubes" in row 1 even these "sphere" positions never be seen during training. 3) Finally, we train a color EBM in combination with both position and shape EBMs to generate images of different shapes at different positions and colors. Again we fix both position and shape EBMs, and only train the color model. In Figure 8 row 3, the objects with different color have the same position as row 1 and same shape as row 2 which shows the EBM can continually learn different concepts and extrapolate new concepts in combination with previously learned concepts to generate new images.

In Table 2, we quantitatively evaluate the continuous learning ability of our EBM and GAN [21]. Similar to the quantitative evaluation in Section 3.2, we a train three classifiers for position, shape, color respectively. For fair comparison, the GAN model is also trained sequentially on the position, shape, and color datasets (with the corresponding position, shape, color and other random attributes set to match the training in EBMs).

The position accuracy of EBM does not drop significantly when continually learning new concepts (shape and color) which shows our EBM is able to extrapolate earlier learned concepts by combining them with newly learned concepts. In contrast, while the GAN model is able to learn the attributes of position, shape and color models given the corresponding dataset. We find the accuracies of position and shape drops significantly after learning color. The bad performance shows that GANs cannot com-

Table 2: Quantitative evaluation of continual learning. A position EBM is first trained on "purple" "cubes" at different positions. A shape EBM is then trained on different "purple" shapes. Finally, a color EBM is trained on shapes of many colors with Earlier EBMs are fixed and combined with new EBMs. We compare with a GAN model [21] which is also trained on the same position, shape and color dataset. EBMs is better at continually learning new concepts and remember the old concepts. (Acc: accuracy)

| Model | Position Acc | Shape Acc | Color Acc |
|---|---|---|---|
| EBM (Position) | 0.901 | - | - |
| EBM (Position + Shape) | 0.813 | 0.743 | - |
| EBM (Position + Shape + Color) | 0.781 | 0.703 | 0.521 |
| GAN (Position) | 0.941 | - | - |
| GAN (Position + Shape) | 0.111 | 0.977 | - |
| GAN (Position + Shape + Color) | 0.117 | 0.476 | 0.984 |

bine the newly learned attributes with
the previous attributes.

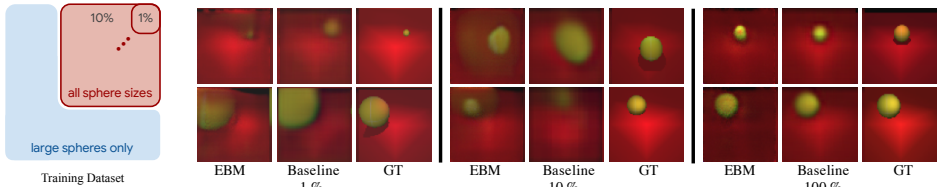

Figure 9: Cross product extrapolation. Left: the spheres of all sizes only appear in the top right corner (1%, 10%, . . . ) of the scene and the remaining positions only have large size spheres. Right: generated images of novel size and position combinations using EBM and the baseline model.

## 4.4   Cross Product Extrapolation

Humans are endowed with the ability to extrapolate novel concept combinations when only a limited number of combinations were originally observed. For example, despite never having seen a "purple cube", a human can compose what it looks like based on the previously observation of "red cube" and "purple sphere".

To evaluate the extrapolation ability of EBMs, we construct a dataset of MuJoCo scene images with spheres of all possible sizes appearing only in the top right corner of the scene and spheres of only large size appearing in the remaining positions. The left figure in Figure 9 shows a qualitative illustration. For the spheres only in the top right corner of the scene, we design different settings. For example, 1% meaning only 1% of positions (starting from the top right corner) that contain all sphere sizes are used for training. At test time, we evaluate the generation of spheres of all sizes at positions that are not seen during the training time. Similar to 1%, 10% and 100% mean the spheres of all sizes appears only in the top right 10% and 100% of the scene. The task is to test the quality of generated objects with unseen size and position combinations. This requires the model to extrapolate the learned position and size concepts in novel combinations.

We train two EBMs on this dataset. One is conditioned on the position latent and trained only on large sizes and another is conditioned on the size latent and trained at the aforementioned percentage of positions. Conjunction of the two EBMs is fine-tuned for generation through gradient descent. We compare this composed model with a baseline holistic model conditioned on both position and size jointly. The baseline is trained on the same position and size combinations and optimized directly from the Mean Squared Error between the generated image and real image. Both models use the same architecture and number of parameters are described in the appendix.

We qualitatively compare the EBM and baseline in Figure 9. When sphere of all sizes are only distributed in the 1% of possible locations, both the EBM and baseline have bad performance. This is because the very few combinations of sizes and positions make both models fail in extrapolation. For the 10% setting, our EBM is better than baseline. EBM is able to combine concepts to form images from few combination examples by learning an independent model for each concept factor. Both EBM and baseline models generate accurate images when given examples of all combinations (100% setting), but our EBM is closer to ground truth than the baseline.

In Figure 10, we quantitatively evaluate the extrapolation ability of EBM and the baseline. We train a regression model that outputs both the position and size of a generated sphere image. We compute the error between the predicted size and ground truth size and report it in the first image of Figure 10. Similarly, we report the position error in the second image. EBMs are able to extrapolate both position and size better than the baseline model with smaller errors. The size errors goes down with more examples of all sphere sizes. For position error, both EBM and the baseline model have smaller errors at 1% data than 5% or 10% data. This result is due to the make-up of the data – with 1% data, only 1% of the rightmost sphere positions have different size annotations, so the models generate large spheres at the conditioned position which are closer to the ground truth position since most positions (99%) are large spheres.

## 4.5   Concept Inference

Our formulation also allows us to infer concept parameters given a compositional relationship in inputs. For example, given a generated set of of images, each generated by the same underlying concept (conjunction), the likelihood of a concept is given by:

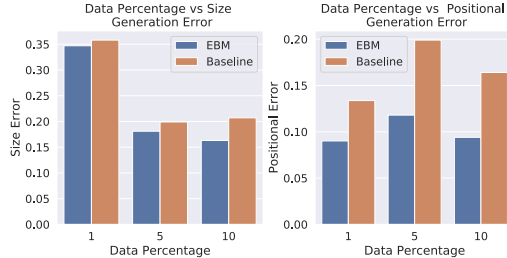

Figure 10: Cross product extrapolation results with respect to the percentages of areas on the top right corner. EBM has lower size and position errors which means EBM is able to extrapolate better with less data than the baseline model.

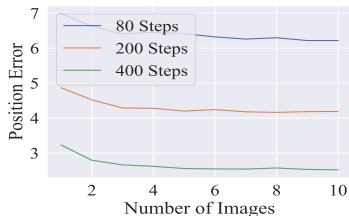

Figure 11: Concept inference from multiple observations. Multiple images are generated under different size, shape, camera view points, and lighting conditions. The position prediction error decreases when the number of input images increases with different Langevin Dynamics sampling steps for training.

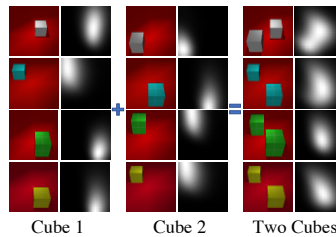

Figure 12: Concept inference of multiple objects with EBM trained on single cubes and tested on two cubes. The color images are the input and the gray images are the output energy map over all positions. The energy map of two cubes correctly shows the bimodality which is close to the summation of the front two energy maps.

$$p(x_1, x_2, \ldots, x_n | c) \propto e^{-\sum_i E(x_i | c)}. \tag{11}$$

We can then obtain maximum a posteriori (MAP) estimates of concept parameters by minimizing the logarithm of the above expression. We evaluate inference on an EBM trained on object position, which takes an image and an object position (x,y in 2D) as input and outputs an energy. We analyze the accuracy of such inference in the appendix and find EBMs exhibit both high accuracy and robustness, performing before than a ResNet.

**Concept Inference from Multiple Observations** The composition rules in Section 3.2 apply directly to inference. When given several different views of an object at a particular position with different size, shape, camera view points, and lighting conditions, we can formulate concept inference as inference over a conjunction of multiple positional EBMs. Each positional EBM takes a different view as input we minimize energy value over positions across the sum of the energies. We use the same metric used above, i.e. Mean Absolute Error, in position inference and find the error in regressing positions goes down when successively giving more images in Figure 11.

**Concept Inference of Unseen Scene with Multiple Objects** We also investigate the inherent compositionality that emerges from inference on a single EBM generalizing to multiple objects. Given EBMs trained on images of a single object, we test on images with multiple objects (not seen in training). In Figure 12, we plot the input RGB image and the generated energy maps over all positions in the scene. The "Two Cubes" scenes are never seen during training, but the output energy map is still make scene with the bimodality energy distribution. The generated energy map of "Two Cubes" is also close to the summation of energy maps of "Cube 1" and "Cube 2" which shows the EBM is able to infer concepts, such as position, on unseen scene with multiple objects.

## 5 Conclusion

In this paper, we demonstrate the potential of EBMs for both compositional generation and inference. We show that EBMs support composition on both the factor and object level, unifying different perspectives of compositionality and can recursively combine with each other. We further showcase how this composition can be applied to both continually learn and compositionally infer underlying concepts. We hope our results inspire future work in this direction.

# 6 Acknowledgement

We should like to thank Jiayuan Mao for reading and providing feedback on the paper and both Josh Tenenbaum and Jiayuan Mao for helpful feedback on the paper.

# 7 Broader Impacts

We believe that compositionality is a crucial component of next generation AI systems. Compositionality enables system to synthesize and combine knowledge from different domains to tackle the problem in hand. Our proposed method is step towards more composable deep learning models. A truly compositional system has many positive societal benefits, potentially enabling a intelligent and flexible robots that can selectively recruit different skills learned for the task on hand, or super-human synthesis of scientific knowledge that can further progress of scientific discovery. At the same time, there remain unanswered ethical problems about any such next generation AI system.

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
