[Supplementary Material · supplement.pdf]

## A.1 Appendix

### A.1.1 Inference

To evaluate the inference ability of EBMs, we generate a new MuJoCo Scene dataset for training and testing. Each scene has varying lighting conditions with one object, either sphere or cube, at all possible positions and some sizes. We build several different test datasets to evaluate generalization if models. The easiest one is *"Test"* which has the same data distribution with the training dataset. The *"Size"* test dataset contains objects twice the size of training objects. *"Color"* dataset has object colors never been seen during training. *"Light"* is a test dataset with different light sources and *"Type"* dataset consists of cylinder images while the training images are only spheres or cubes.

We evaluate inference on an EBM trained on object position, which takes an image and an object position (x,y in 2D) as input and outputs an energy. We iterate densely over all positions (20 by 20 grid of positions) and select the position with the minimal energy as our inference result. We evaluate this result by computing the Mean Absolute Error between the predicted position and ground truth object position.

We compare EBMs with two baseline models, ResNet model (He et al., 2016) (with the same architecture as EBM) and PixelCNN (Oord et al., 2016). Table 1 shows the comparison results using different number of Langevin Dynamics sampling steps ($k$ in Equation 3 in the main text). We find that inference in EBMs is able to generalize well to different out of distribution datasets such as Color, Light, Size and Type. A large number of Langevin sampling steps also improves performance, with a large number of steps of training exhibiting both better training accuracy and generalization performance.

Table 1: Position error on different test datasets. "Test" has the same data distribution with training set. Other datasets change one environmental parameter, e.g. color, size, type, and light, which are unseen in the training set. "Avg" is the average error of "Color", "Light", "Size", and "Type". "Steps"indicates the number of sampling steps used to train the EBMs. EBMs are able to generalize better on unseen datasets. Larger number of sampling steps significantly decrease overall EBM error.

| Model | Steps | Color | Light | Size | Type | Avg | Test |
|---|---|---|---|---|---|---|---|
| EBM | 200 | 10.899 | 6.307 | 8.431 | 6.304 | 7.985 | 3.903 |
| EBM | 400 | **4.084** | **4.033** | **6.853** | **3.694** | **4.666** | **2.917** |
| Resnet | - | 20.002 | 5.881 | 10.378 | 6.310 | 10.643 | 3.635 |
| PixelCNN | - | 60.607 | 58.589 | 33.889 | 48.138 | 50.306 | 43.460 |

### A.1.2 Partition Function

We estimate the magnitude of the partition function of an EBM by evaluating the energy it assigns to all data points it is trained on, and plot the resultant histogram of energies. Figure A1 shows that the EBMs we train have similar histograms due to a combination of L2 normalization and spectral normalization. The EBMs we evaluated have different architectures but similar histograms.

Figure A1: Energy histograms of models trained on CelebA smiling (left), CelebA attractive (middle) and pretrained CIFAR-10 model from (Du & Mordatch, 2019) (right). Each EBM we evaluate have different architectures but still have similar histograms.

Specifically, in Figure A1, we compare the energy histogram of a CelebA model trained on either smiling or attractive histograms as well as the CIFAR-10 model from (Du & Mordatch, 2019). We

find that all energy histograms are similar, exhibiting minimum and maximum energies between -0.01 and 0.01. This is true even for the CIFAR-10 model which uses a significantly different dataset and architecture.

### A.1.3 Analysis of Mismatch of Partition Function on Disjunction

In scenarios where partition functions are different, our defined disjunction operator does not fail drastically. If two un-normalized probability distributions have partition function values of $w_1$ and $w_2$ then models will be sampled with proportion $\frac{w_1}{w_1+w_2}$ and $\frac{w_2}{w_1+w_2}$, which is not a dramatic failure in disjunction.

### A.1.4 Disjoint Compositionality Results

We further evaluate compositionality when conditioned factors are mutually disjoint from each other. In particular, we train EBM models on frog and truck image classes in CIFAR-10. In Figure A2, we illustrate resulting generations. We find that when conditioning on both classes, our resultant generations exhibit characteristics of each individual class.

Frog

Truck

Frog + Truck

Figure A2: Hybrid combinations of frog and truck EBMs.

### A.1.5 Discussion on Other Generative Models

To sample from the conjunction/disjunction/negation of seperate probability distributions, MCMC must be run. Other generative models, such as autoregressive models, can also support MCMC, but we find that in practice other generative models do not sample well under gradient based MCMC.

(a) Samples Generated from Langevin Sampling on PixelCNN++ model from (Salimans et al., 2017).

(b) Samples Generated from Autoregressive Sampling on PixelCNN++ model from (Salimans et al., 2017).

Figure A3: Comparison on samples generated from different sampling scenes on PixelCNN++ model from (Salimans et al., 2017). We note that Langevin sampling, while not making realistic samples, generate **higher** likelihood samples than those from autoregressive sampling.

We considered Langevin based sampling on the pretrained CIFAR-10 unconditional PixelCNN++ model (Salimans et al., 2017) in Figure A3. While both sampling schemes generate images with similar likelihoods (with Langevin sampling creating higher likelihood samples), we find images generated from Langevin sampling are significantly worse than those generated from autoregressive sampling. We speculate that EBMs fit the MCMC sampling procedure better than other models since EBMs are trained with MCMC inference, and are thus less susceptible to adversarial modes.

### A.1.6 Models

| 3x3 conv2d, 64 |
| --- |
| ResBlock down 64 |
| ResBlock down 128 |
| ResBlock down 128 |
| ResBlock down 256 |
| Global Mean Pooling |
| Dense → 1 |

(a) The model architecture of EBM used on the Mujoco Scenes Dataset.

| Dense → 4096 |
| --- |
| Reshape → 256x4x4 ResBlock up 256 |
| ResBlock up 128 |
| ResBlock up 64 |
| ResBlock up 64 |
| 3x3 conv2d, 3 |

(b) The model architecture of baseline model for joint generation (section A.1.2).

| 3x3 conv2d, 64 |
| --- |
| ResBlock down 64 |
| ResBlock down 128 |
| ResBlock down 256 |
| ResBlock down 512 |
| ResBlock down 1024 |
| ResBlock 1024 |
| Global Sum Pooling |
| dense → 1 |

(c) The model architecture of EBM used on the CelebA Dataset.

Figure A4: Architecture of models on different datasets.

We detail the EBM architectures used for the Mujoco Scenes images in Figure A4a and for the Celeba 128x128 images in Figure A4c. The baseline model used for comparisons in section 3.4 is in Figure A4b.

### A.1.7 Training Details/Hyperparameters/Source Code

Models trained on Mujoco Scenes and CelebA datasets use the Adam optimizer with the learning rate 3e-4, first order moment 0.0, and second order moment 0.999. The batch size is 128. The replay buffer size is 50000 with a 5% replacement rate. Spectral normalization is applied to models with a step size of 100 for each Langevin dynamics step. We use 60 steps of Langevin sampling per training iteration for the CelebA dataset and 80 steps of Langevin sampling per training iteration for the Mujoco Scenes dataset. We use the Swish activation to train our models (as noted in (Du & Mordatch, 2019)), and find that it greatly stabilizes and speeds up training of models.