[Reviews · NeurIPS 2020]

Review 1

Summary and Contributions: The paper presents an approach that potentially can provide a long search for concept in machine learning - posterior concept compositionallity. The authors propose a framework in which a "concept" (or a feature, such as position, color, hair color, gender, geometric shape etc.) is learned from a set of images where all other "concepts" are kept fixed. After this is done to each concept separately, an image that portrays a requested logical combination of the concepts is produced, without having the system see any combined result. This is, in a sense, a generalization of disentanglement, where a single factor is learned to be extracted away from all others typically.

Strengths: The demonstrated achieved capability is an interesting one, which most probably can bear merit for the community.

Weaknesses: - The quality of the results are underwhelming, but this is to be expected since the system has never seen composed examples. - Some of the details are still somewhat unclear.

Correctness: I am not an expert, but everything seems sound to me

Clarity: The paper is generally well written and presents the main ideas in a coherent and intuitive manner. Some of the vital details are missing though, especially regarding the actual generation step, once the concepts have been composed and an X has been decided. Some experimentation regard the quality of output as it depends on some of the choices in this last step is interesting to see, especially in order to help understand how the method behaves.

Relation to Prior Work: Seems decent. Some more of the disentanglement work can be mentioned.

Reproducibility: Yes

Additional Feedback: After reading the rebuttal and other reviews, my opinion of this work did not change. Even though the generation quality is not great (a concern that was well addressed by the rebuttal), I think the new capabilities the EBM model exhibits are exciting, and incredibly useful, and different compared to anything we have seen before, hence the degradation of quality is not concerning. I do, however, completely agree we everything Reviewer #2 wrote in answer 8, and think these improvements should be incorporated into the final version.


Review 2

Summary and Contributions: The paper explores compositionality in generative models and discusses how to use energy models in order to achieve compositionality of objects or properties. It explores the model in the contexts of faces (CelebA) and artificial block images with a couple of objects and multiple properties that can be controlled such as shape, position and color. Update: I thank the authors for responding to the review. I was satisfied with the responses to my questions and potential concerns and happy to improve my score. It would be great to update the final version of the paper accordingly based on the author's response. Best of luck!

Strengths: * Developing a new energy based model for generating natural images. * Ability to perform controlled generation that combined different properties through logical operations. * Compositional generalization to new unseen combinations of concepts.

Weaknesses: * The visual quality/fidelity of the generated images is quite low. Making sure that the visual fidelity on common metrics such as FID matches or is at least close enough to GAN models will be useful to validate that the approach supports high fidelity (as otherwise it may be the case that it achieves compositionality at the expense of lower potential for fine details or high fidelity, as is the case in e.g. VAEs). Given that there have been many works that explore combinations of properties for CelebA images with GANs, showing that the proposed approach can compete with them is especially important. * It is unclear to me if MCMC is efficient in terms of training and convergence. Showing learning plots as well compared to other types of generative models will be useful. * The use of energy models for image generation is much more unexplored compared to GANs and VAEs and so exploring it further is great. However, note that the motivation and goals of the model -- to achieve compositional generation through logical combination of concepts learned through data subsets, is similar to a prior VAE paper. See further details in the related work review part. * Given the visual samples in the paper, it looks as if it might be the case that the model has limited variability in generated images: the face images in figure 3 show that both in the second and 4th rows the model tends to generate images that feature unspecified but correlated properties, such as the blonde hair or the very similar bottom three faces. That’s also the case in figure 5 rows 2-4. Consequently, it gives the sense that the model or sampling may not allow for large variation in the generated images, but rather tend to take typical likely examples, as happened in the earlier GAN models. A quantitative comparison of the variance in the images compared to other types of generative models will be useful to either refute or validate this.

Correctness: The claims in the paper are corroborated by experiments and the model description is clear and correct.

Clarity: The paper clarity is good and it is well written and easy to follow.

Relation to Prior Work: The paper does a good job in comparing to other approaches both of the same family of generative models and models of other types. However, there is a particular paper SCAN: Learning Hierarchical Compositional Visual Concepts that has very similar motivation, and also shows a model that can learn concepts through subsets of the data and combines them together through logical operations. They also show that their model can achieve through that generalization to unseen combinations. So the high-level motivation is shared with that paper, even though the means to achieve that are different (they use VAEs rather than energy models). Comparing and contrasting these approaches in the paper will be useful. In that regard, it will be useful to revise the claim about the “approach is the only approach enabling such compositionality across independently trained models.”.

Reproducibility: Yes

Additional Feedback: * The correspondence between the logical operations and equations in the first figure is unclear to me and so would be useful to add explanation about it. * The authors mention the model supports generalization to unseen combinations of concepts, and that concepts are defined by training over particular subsets of images. What happens if concepts are merged together for image subsets that do not naturally overlap (e.g. cats and dogs). Will the model learn to generate hybrids of them, that blend together some of the properties of one class with other properties of the other class, or will fail due to the gap between the two distributions that do not overlap? * The paper mentions that attention-based approaches towards compositionality “may have difficulty capturing the global structure of an image”. Why is that the case? I would presume that actually such models will have advantage in capturing the global image structure, since they can decompose it into parts while other methods have to account for all possible object rearrangements through one model that can’t explicitly attend or isolate particular regions. * It would be useful to add a remark below the first figure that the equations will be explained in page 4 of the paper. * In figure 10 I would expect actually a much clearer bimodal separation that should ideally look almost like the sum of the two energy maps for each object. Except the first row and maybe the second, the results in that figure look to me a bit negative. * The generalization results in the appendix are nice and may be useful to add some of the into the main paper if space permits. * Merging section 3.3 with the experimental section will be more useful. Otherwise there is a description of potential applications without any concrete following examples or results. Small comments: * In line 166: ”of of” duplication.


Review 3

Summary and Contributions: This paper considers training a set of independent energy-based models (EBMs) for different concepts and compose them based on three different logical operators over concepts. This is a new approach to compositionality with separate EBMs compared to previous methods cited in the paper. The proposed method enables interesting applications such as continual learning with reusing past EBMs for image generation so that different compositions of concepts can be present in the generated images as the continual learning progresses. This paper provides both qualitative and quantitative evaluations for compositional concept image generation tasks in Mujoco Scene domain and CelebA domain to justify their three methods of combining energy functions. Concepts extrapolation, continual learning and concept inference are also evaluated in Mujoco Scene domain.

Strengths: The simple derivations on the equivalence between different ways of combining energy functions and different logical operators is neat and interesting. (though product of mixture is proposed before and the simplified expression for disjunction assumes equal partition function for different models) This is the first work to actually generate realistic images with composite concepts using Langevin dynamics. The performance on continual learning is also promising with better quantitative results than GANs.

Weaknesses: The paper claims L2 normalization and spectral normalization can make partition function behave similarly for different models trained on same (but with different concepts) or different datasets but I am not sure whether this conjecture is true. Is there any ablation studies (e.g. by removing spectral normalization) to support this claim? It might just make training unstable by removing it but this can still be checked and mentioned in the paper. I believe you could also make some simple synthetic toy data to expand on the conditions when this argument holds.

Correctness: Besides the equal partition function assumption and lack of ablation studies to check verify that claim, I find the empirical components of the paper to be good with various qualitative results on the successful composition results, and also covers an quantitative evaluation by using a good supervised classifier for MuJoCo Scnene as the critic, for example.

Clarity: Yes, this paper is overall well written.

Relation to Prior Work: Yes, the contributions of this paper are clearly stated in Section1. The difference is also clearly stated.

Reproducibility: Yes

Additional Feedback: Training multiple energy models seem to be computationally challenging and might not scale in practice. Is there any comments on how this can be dealt with the proposed method? ----------------- Update after authors' response------------ Thank you for the clarifications. With the inclusion of these clarifications, I think the paper is quite clear about its assumption and experimental protocols for verifying those assumptions. I will update my score to 7


Review 4

Summary and Contributions: In this paper, the authors firstly address human capabilities that humans can combine a finite number of primitive components and compose increasingly complex concepts using those components by recombining and reusing in novel ways. They are interested in enabling such capabilities in machine learning systems, particularly in the context of generative modeling. Thus, in this paper, the authors propose to implement the compositionality via energy based models (EBMs). Instead of an explicit vector of factors that is input to a generator function or object slots that are blended to form an image, the proposed method defines factors of variation and object slots via energy functions. The contribution of this paper is threefold: (1) the authors show that composition of energy-based model can be used to generate plausible natural images, while previous works have shown in abstract setting, (2) they propose a principled approach to combine independent trained energy models based on logical operators (i.e., conjunction, disjunction, and negation) which can be chained recursively, allowing controllable generation, and (3) this paper allows to extrapolate to new concept combinations, continually incorporate new visual concepts for generation, and infer concept properties compositionally.

Strengths: 1) To the best of my knowledge, this is the first work in demonstrating the potential of EBMs for both compositional generation and inference. While composition of energy-based models has been proposed in abstract settings before, this paper newly shows that it can be used to generate natural images. 2) Thorough experiments regarding contributions are well conducted. This paper clearly provides both quantitative results and qualitative visualizations in compositional generation, continual learning, and concept inference.

Weaknesses: 1) While the proposed idea is novel, the overall proposed method seems to be the adaptation from the previous works [1,2], thus seemingly incremental. In addition,thorough explanation of energy based models (EBMs) is not sufficient. Preliminary explanation of EBMs is necessary. 2) There is no qualitative comparison between this work and previous works, such as generative adversarial network. It is better to shown qualitative comparison between EBMs and other generative models related to your work. Moreover, in looking at the qualitative result on the proposed method, the generated images seem to be not promising, showing blurry and not detailed facial images. [1] G. E. Hinton. Products of experts. International Conference on Artificial Neural Networks, 1999. [2] Y. Du and I. Mordatch. Implicit generation and generalization in energy-based models. arXiv preprint 318 arXiv:1903.08689, 2019. [3] M. Welling and Y. W. Teh. Bayesian learning via stochastic gradient langevin dynamics. In Proceedings of 374 the 28th International Conference on Machine Learning (ICML-11), pages 681–688, 2011.

Correctness: Overall, the empirical methodology seems correct. Some typos regarding equations are addressed in additional comments section.

Clarity: Generally, this paper is well written and easily readable. However, I do not think this paper is fully self-contained. Several mathematical equations are not fully understandable in reading this paper only. In equation 2, I understand that this equation is exactly mentioned in [1], I think that there should be more comprehensive explanation (e.g., what are x+ and x-). There should be careful look for mathematical notations. Few additional comments about this paper overall is mentioned in additional feedback section. [1] Du, Yilun, and Igor Mordatch. "Implicit generation and generalization in energy-based models." arXiv preprint arXiv:1903.08689 (2019).

Relation to Prior Work: In this paper, the authors apply energy based models (EBMs) to the task of compositional generation. This paper clearly discusses about difference between itself and previous works in related work section. The authors mention that their approach towards compositionality focuses on composing independently learned probability distribution concepts, while previous works rely on joint training to learn compositions of a fixed number of concepts when utilizing EBMs.

Reproducibility: Yes

Additional Feedback: Here are the additional comments regarding the overall paper: 1) Overall mathematical equations are not kindly explained. The partition function, Z(c_i) in line 127 should be more elaborated. 2) There seems to be an error in equation 7 and 8, regarding logsumexp(-E(x|c_i)). I think the correct writing should be logsumexp(-E(x|c_1), -E(x|c_2), ..., -E(x|c_i)), or the authors can make another equation that defines the most right part of equation 7. Please check. 3) Additional explanations for figures are needed. For example, please explain each column of Figure 8. 4) I understand that there is limited space for writing a paper. However, the formatting instruction says that all tables must be centered, but Table 1 and Table 2 do not seem to be located correctly. 5) Few grammatical errors are appeared: a) In line 74, can not -> cannot b) In line 216, the conjunction of EMBs are ... -> the conjunction of EMBs is ... c) Please try to avoid using the contracted form in formal paper: in line 264, doesn't -> does not. I would also like to see the feedbacks from the authors.

[Author Response · NeurIPS 2020]

We thank reviewers for their valuable comments. We will incorporate all feedback in our final manuscript.

**Sample Quality/Qualitative/Quantitative Comparisons (R1, R2, R4)**   R1/R2/R4 comment on the overall genera-
tion quality of EBMs. We note that the main focus of our paper is to introduce a set of compositional logical operators
over EBMs, and empirical applications, with qualitative fidelity of generations an orthogonal direction. By training an
EBM with a larger number of parameters and computational resources, we see in Figure 1 that EBMs achieve high
fidelity composition results comparable to those of a GAN model (SNGAN is a 128x128 model specifically trained
with Young/Female/Smiling/Wavy Hair attributes). The SNGAN model has the same number of parameters and is
trained with the same number of training iterations using the Mimicry * GAN library. We will release the code for
training these models and pretrained weights.

In terms of image fidelity, on the Young AND Female
AND Smiling AND Wavy Hair split, our composed EBM
obtains an FID of 45.3 while SNGAN obtains FID scores
of 74.2 (all FIDs are large due to a small dataset). R2 fur-
ther asks diversity evaluation. We compare standard devi-
ation across pixels of generated images and find SNGAN
obtains 55.4 while EBMs obtain 64.5, providing evidence
EBMs generate more diverse samples.

Figure 1: High resolution samples of attribute compositionality
with EBMs (same setup as Figure 3 in the main paper). The last row
shows SNGAN samples trained on specific attribute combination.

**Training Time Comparisons with Other Models (R2)**
A general comparison between training EBMs and other
generative models can be found in the appendix A.5 of [1].
Our EBM models are trained with the same methodology,
and exhibit similar trends. EBMs are slower to train
than GAN models, but faster to train than autogressive
and flow models. In the particular setting of qualitative
generation in Figure 1, EBM models roughly take 20
times longer to train than the corresponding SNGAN
model (due to generating negative samples).

**Relations to Previous Work (R2/R4)**   SCAN learns a
fixed latent space for concepts and composition of con-
cepts via logical rules is achieved by manipulating this
latent space. Extending the space of concepts requires retraining the network. With our work, we investigate an
alternative approach where new concepts can be added on demand via new energy functions without invalidating
previous energy functions. This unique characteristic allows for unique benefits – such as the ability to learn visual
concepts in a continual manner. Different from past work in EBMs, our approach is the first to propose additional logical
operators of disjunction and negation, and show that these logical operators can be composed and nested together.

**Compositions of Non-Overlapping Concepts (R2)**   We train seperate EBMs on the frogs
and trucks CIFAR-10 images. We combine models in Figure 2, and find somewhat reasonable
generations that share properties in both classes, although we do not expect good results in
this regime generally.

**Comparison to Attention Masks (R2)**   While image masking approaches to composi-
tionality enable a part-like decomposition of a scene, generation of each part is largely
independent. This can miss interaction effects between parts (such as shadow casting). Our
EBM composition generates all pixels of the image jointly, offering potential of capturing
such interaction effects.

Figure 2: Hybrid combi-
nations of frog and truck
EBMs.

**Equality of Partition Functions (R3)**   We had difficulty in checking for equality of par-
tition functions without using spectral or L2 regularization as they are necessary for stable
training of our method. We will clarify this in the paper.

**Multiple Energy Functions (R3)**   Ensuring that individual energy functions all have good generative performance
can be difficult. We find that using our proposed operators to compose generation can lessen the need for any individual
model to have good generative performance (see for example Figure 1).

**Additional Feedback/Clarification/Typos (R2, R4)**   We will add remarks in figure 1, and merge 3.3 with experiments
section and the generalization. We will add a longer introduction about the EBMs and early equations. We will further
fix the typos from R2 and R4, including the Logsumexp expression.

## Footnotes

*https://github.com/kwotsin/mimicry


[Meta-Review · NeurIPS 2020]

While the notions (energy models, product of experts) and leveraged algorithms (Langevin MCMC, Contrastive Divergence training) are not novel, all reviewers (and the AC) appreciated this work as the first to investigate and demonstrate the potential of EBMs for compositional generation from independently learned probability distribution concepts, successfully on real images. Authors responded well to the reviewers questions and suggestions, providing convincing additional experimental results, incl. higher quality generations standing the comparison with GANs. Please incorporate these in the final version of the paper as well as all the useful clarifications provided in your response to reviewers. With this, the AC recommends the paper be accepted at NeurIPS.